# Adaptive Granularity Graph Rewiring via Granular-ball for Graph Clustering

## Abstract

Graph clustering aims to partition a graph into homogeneous groups of nodes, capturing the graph's node features and connectivity structure. Graph neural network-based approaches excel in node clustering by leveraging the homophilic assumption, which posits that neighboring nodes are likely to share similar characteristics, but low-homophily edges can introduce noise, potentially reducing clustering accuracy. Previous work rewires connections by estimating homophily at an overly fine granular, primarily based on the similarity of connected nodes. Nevertheless, they largely neglect the fact that homophily is distributed across multiple granular levels within the graph. Considering the multi-granular nature of the homophily distribution, we could better differentiate between homophilic and heterophilic nodes at the optimal granularity. To this end, we propose a novel Adaptive Granular Graph Rewiring method (AGGR) that adaptively identifies homophilic regions at appropriate granularities and subtly enhances homophily within the graph structure through graph rewiring, significantly improving GNN performance and clustering outcomes. Specifically, AGGR introduces an Adaptive Granular-Ball graph refinement mechanism to capture homophilic structures within graphs. In addition, a Multi-Granularity Graph Rewiring method is further proposed to add highly homophilic social relations intra-homophilic domains and cut low homophilic relations inter-them. Moreover, we propose a Multi-Task Homophily Refinement Learning framework to integrate the optimization of graph rewiring with graph clustering. Extensive experiments conducted on benchmark datasets demonstrate that AGGR outperforms the state-of-the-art method.

## 1 Introduction

Graph clustering Schaeffer (2007) seeks to divide a graph into homophily groups of nodes, facilitating a deeper understanding of its structural organization. By uncovering such patterns, these algorithms play a crucial role in applications like anomaly detection Ahmed et al. (2021), molecular mining Grunig et al. (2022), social network analysis Nettleton (2013), and beyond.

Capturing the graph's node features and connectivity structure is the main theme for graph clustering. Early work Girvan & Newman (2002); Hastings (2006) on graph clustering primarily focused on extracting shallow features from the graph, without incorporating deep representation learning, and often underestimated the complex interplay between node features and structural information. Recent advancements in Graph Neural Networks (GNNs) Liu et al. (2022a); Yue et al. (2022); Gong et al. (2022) have significantly enhanced graph clustering by enabling the extraction of informative representations that effectively integrate the graph's topological structure with the rich attributes associated with its nodes. Based on the *homophily* assumption on the neighboring nodes, GNNs recursively learn node embeddings through the aggregation of information from connected nodes. While GNN-based graph clustering methods have achieved significant progress, their reliance on the raw graph structure makes them susceptible to noise from low-homophily edges, which can compromise clustering accuracy. To address this issue, HoLe Gu et al. (2023) improves graph quality by estimating homophily through node similarity, removing low-homophily edges, and adding high-homophily edges. However, it relies on overly fine-grained homophily estimation and overlooks the multi-granularity of homophilic structures within the graph.

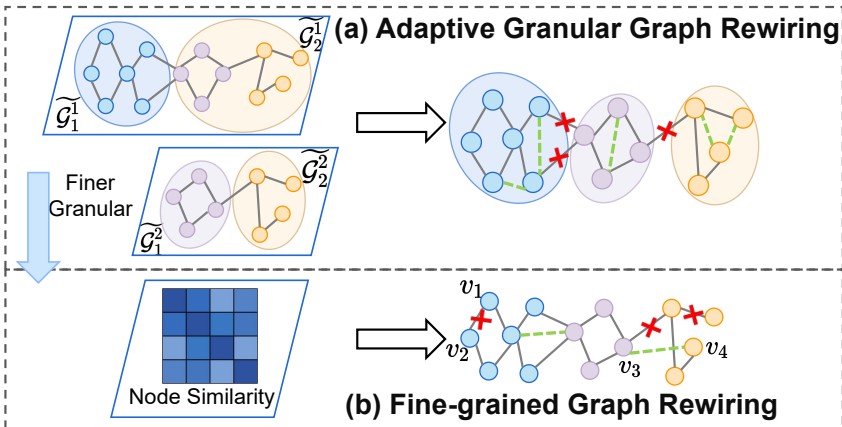

Figure 1: The comparison of Adaptive Granularity Graph Rewiring and Fine-grained Graph Rewiring.

In real-world graph clustering scenarios, graphs are inherently composed of homophily subregions that exhibit varying levels of granularity, where nodes within each subregion demonstrate strong homophilic properties. As illustrated in Fig. 1 (a), the graph as a whole can be considered the coarsest granularity of homophilic structure, with progressively finer-grained levels of structure represented from top to bottom. At the finest granularity (as illustrated in Fig. 1 b), homophily is defined at the level of individual edges, quantified by the similarity between the connected nodes. Each node in the graph is associated with an adaptive-grained homophilic subdomain, tailored to its specific structural context. For the example in Fig. 1 (a), $\widetilde{\mathcal{G}}_1^1$ is at the appropriate granularity, while $\widetilde{\mathcal{G}}_2^1$ should be further decomposed into $\widetilde{\mathcal{G}}_1^2$ and $\widetilde{\mathcal{G}}_2^2$ at a finer granularity. Ignoring the adaptive granularity of homophily (as illustrated in Fig. 1 b), nodes with strong topological correlations but low similarity ($v_1$ and $v_2$) may be incorrectly judged as low homophily. Moreover, connections might be mistakenly established between similar nodes ($v_3$ and $v_4$) that do not share meaningful correlations within the broader structure of the graph. Considering the multi-granularity distribution of homophily allows for a more comprehensive understanding of the graph's diverse structural and relational patterns, which enables more precise identification of meaningful connections while reducing the influence of noisy or spurious edges.

Actually, it is non-trivial to model the multi-granularity distribution of homophily within the graph for graph rewiring in graph clustering, due to three challenges: 1) Homophilic structure refinement: Homophilic structures are organized by nodes with high connectivity and strong feature similarity. These structures should be modeled at the appropriate granularity, meaning that no coarser-grained parent graph or finer-grained subgraph should exhibit a higher homophily ratio than the granularity at which the structure is being analyzed. Inspired by the advantage of granular-ball computingXia (2019); Xie et al. (2023) in modeling the multi-granular characteristics for scatter data, we further explore its potential for graph refinement and propose an Adaptive Granular-Ball graph refinement mechanism to capture homophilic structures within the graph. 2) Differentiating homophilic and heterophilic edges for graph rewiring: Homophilic subdomains facilitate homophily assessment by leveraging the graph's structural properties, while node similarity offers an additional perspective for quantifying homophily. Therefore, it is essential to develop a graph rewiring method that incorporates both factors. 3) Joint Optimization of graph rewiring with clustering: Since both graph rewiring and graph clustering are typically unsupervised tasks, directly using clustering results to guide the refinement of the graph structure is challenging. Therefore, a more refined approach is needed to selectively leverage reliable clustering outcomes, ensuring they effectively inform the graph rewiring process and ultimately improve clustering accuracy.

To address these challenges, we propose a novel Adaptive Granular Graph Rewiring method (AGGR). AGGR introduces an Adaptive Granular-Ball Graph Refinement mechanism to capture homophilic structures at the optimal granularity. In addition, a Multi-Granularity Graph Rewiring approach is introduced, which enhances homophilic relationships within homophilic regions and re-

moves low-homophily connections between distinct homophilic structures. Furthermore, we present a Multi-Task Homophily Refinement Learning framework to effectively utilize intermediate clustering results, facilitating structure refinement throughout the training process. Extensive experiments conducted on seven real-world datasets demonstrate that AGGR consistently outperforms state-of-the-art methods.

In a nutshell, this work makes the following contributions:

- We emphasize the role of multi-granularity homophily distribution for graph clustering.

- We propose a novel AGGR, that adaptively refines the homophilic structures within graphs, and subtly enhances homophily within the graph structure. A Multi-Task Homophily Refinement Learning framework is further proposed to integrate the optimization of graph rewiring with graph clustering.

- We conducted extensive experiments on seven real-world datasets and AGGR effectively improves the performance of graph clustering.

## 2 RELATED WORK

Graph clustering aims to partition a graph into homophily groups of nodes, leveraging both node features and the graph's connectivity structure. Traditional approaches often relied on shallow methods to analyze graph structures. For instance, centrality indices were used in Girvan & Newman (2002) to identify community boundaries, while belief propagation was applied in Hastings (2006) for uncovering social groups. Although these methods addressed basic relationships between graph structure and node features, they largely overlooked the complex interaction between them.

With the advent of deep learning, significant advancements have been achieved in graph node clustering through the use of deep architectures Cai et al. (2022a;b); Zhao et al. (2023). Among the pioneering deep clustering models, autoencoders Hinton & Salakhutdinov (2006); Xie et al. (2016); Guo et al. (2017); Wang et al. (2016) stood out for their ability to learn low-dimensional feature representations via nonlinear transformations. Despite the progress, these approaches underutilized the graph's structural information, limiting their ability to fully exploit the underlying graph topology Cao et al. (2016); Ghasedi Dizaji et al. (2017).

Graph neural networks (GNNs) excel at jointly modeling node features and structural relationships through message-passing, making them highly effective for tasks like graph clustering, where integrating feature and topology information is crucial Liu et al. (2022a); Yue et al. (2022). In the context of graph clustering, GNN-based methods can be broadly classified into two-stage and one-stage frameworks. Two-stage methods first derive node embeddings in an unsupervised manner and then apply clustering algorithms to the learned embeddings. For example, methods like Liu et al. (2022c); Bo et al. (2020); Tu et al. (2021); Peng et al. (2021) incorporate auxiliary distribution alignment during self-training to refine embeddings. In contrast, one-stage approaches integrate representation learning and clustering into a unified framework, where clustering feedback is directly used to optimize node representations. Representative methods include AGC-DRR Gong et al. (2022), which introduces contrastive constraints within a joint clustering framework, and GCC Fettal et al. (2022), which minimizes the discrepancy between embeddings and cluster centroids in a cohesive learning process. Recent research such as DGClusterBhowmick et al. (2024) optimizes clustering performance using modularity-based objectives, while FPGCXie et al. (2025) introduces a squeeze-and-excitation block to identify cluster-relevant features for each node. Despite effectiveness, their dependence on the raw graph structure fails to address the influence of low-homophily edges, which can introduce noise into node representations and undermine the accuracy of clustering outcomes. To address this issue, HoLe Gu et al. (2023) estimates homophily through node similarity, to remove edges with low homophily and incorporate edges with high homophily. Although HoLe enhances GNN performance in graph clustering by improving graph quality, its reliance on node similarity to evaluate homophily is limited by an excessively fine granularity. This approach disregards the multi-granularity nature of homophilic structures within the graph and remains vulnerable to noise, potentially resulting in inaccurate assessments.

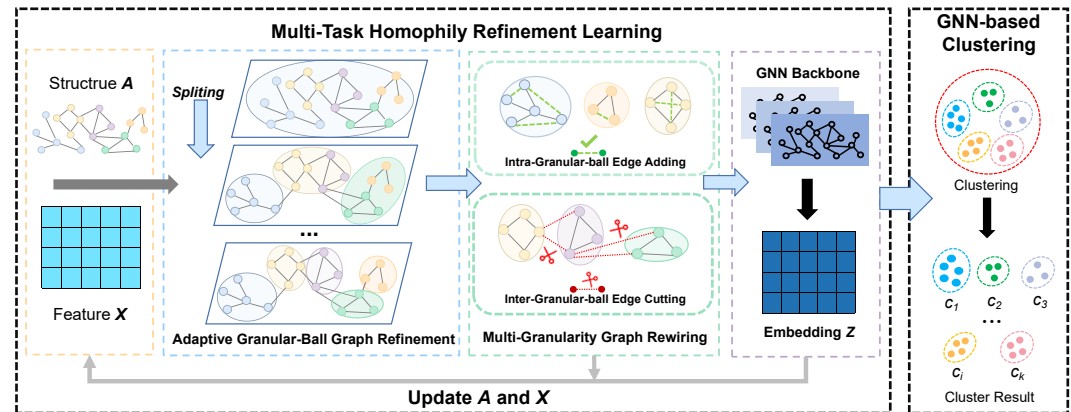

Figure 2: An overview of the proposed AGGR framework.

## 3 METHODOLOGY

We designed Adaptive Granularity Graph Rewiring via Granular-ball for Graph Clustering(AGGR). Fig.2 presents an overview of our proposed AGGR framework. It comprises three distinct modules: (a) Adaptive Granular-Ball Graph Refinement, which iteratively refines the graph into multiple homophily regions. (b) Multi-Granularity Graph Rewiring, which enhances homophily relationships within subregions and eliminates low-homophily connections inter subregions. (c) Multi-Task Homophily Refinement Learning, which refines granular-ball graph rewiring using embeddings and clustering results output by the GNN, enabling iterative optimization between the two modules to enhance performance. We start this section by introducing important notations.

### 3.1 NOTATIONS AND DEFINITIONS

Consider an undirected attribute graph $\mathcal{G} = \{\mathcal{V}, \mathcal{E}, \mathbf{X}, \mathbf{A}\}$, where $\mathcal{V} = \{v_1, v_2, \cdots, v_N\}$ is a set of $N$ nodes, $\mathcal{E}$ denotes the set of edges, and $\mathbf{X} \in \mathbb{R}^{N \times d}$ is the associated feature matrix. Each row of this matrix $\mathbf{x}_i \in \mathbb{R}^d$ denotes the vector associated with the node $v_i$, and $d$ is the dimensionality of node features. The edges are represented by an adjacency matrix $\mathbf{A} = \{a_{ij}\} \in \mathbb{R}^{N \times N}$, where $a_{ij} = 1$ if $(v_i, v_j) \in \mathcal{E}$ and $a_{ij} = 0$ otherwise.

**Definition 1. Granular-ball Computing.** Granular-ball Computing is a computational paradigm defined on a metric space $(\mathcal{G}, d)$, where $\mathcal{G} = (\mathcal{V}, \mathcal{E}, \mathbf{X}, \mathbf{A})$ denotes a structured domain (e.g., a graph) endowed with a distance function $d : \mathcal{V} \times \mathcal{V} \rightarrow \mathbb{R}_{\geq 0}$. The space $\mathcal{G}$ is covered by a finite collection of closed metric balls, each referred to as a Granular-ball:

$$\widetilde{\mathcal{G}}_i = \{v \in \mathcal{V} \mid d(v, c_i) \leq r_{c_i}\}, \tag{1}$$

where $c \in \mathcal{V}$ is a designated center node, and $r_c$ is a radius determined to ensure coverage and local coherence. Each $\widetilde{\mathcal{G}}_i$ defines a basic closed neighborhood in the topology induced by $d$, and the collection $\widetilde{\mathbb{G}} = \bigcup_{i=1}^n \widetilde{\mathcal{G}}_i$ forms a topological covering of $\mathcal{V}$. The metric $d$ may integrate structural and feature-based proximity, enabling localized representation and processing within each Granular-ball. This supports scalable and topology-aware computations over the graph domain. We formalize the Granular-ball optimization problem as follows:

$$\min \frac{1}{n} \sum_{i=1}^n \frac{r_i}{Q(\widetilde{\mathcal{G}}_i^t)} + n \quad \text{s.t. } Q(\widetilde{\mathcal{G}}_i^t) > \frac{1}{n_i} \sum_{j=1}^{n_i} Q(\widetilde{\mathcal{G}_j^{t+1}}), \tag{2}$$

where $Q(\cdot)$ is the quality of $\widetilde{\mathcal{G}}$, $t$ denotes the number of Granular-ball splitting iterations and $n_i$ is the number of Granular-balls resulting from the refinement of $\widetilde{\mathcal{G}}_i^t$. Inspired by modularitynew (2006), the formula for calculating the Granular-ball quality $Q$ is as follows:

$$Q(\widetilde{\mathcal{G}}) = \frac{1}{2m} \sum_{ij} (\mathbf{A}_{ij} - \frac{d_i d_j}{2m}) \cos(\mathbf{x}_i, \mathbf{x}_j), \tag{3}$$

where $\mathbf{A}_{ij}$ is the adjacency matrix of $\widetilde{\mathcal{G}}$. $d_i$ denotes the degrees of node $v_i \in \mathcal{V}_{\widetilde{\mathcal{G}}}$, and $m$ is the total number of edges in $\widetilde{\mathcal{G}}$.

## 3.2 ADAPTIVE GRANULAR-BALL GRAPH REFINEMENT

In this section, we provide a detailed explanation of **the Adaptive Granular-ball Graph Refinement**. As depicted in Fig.2, the original graph $\mathcal{G}$ is initially treated as a single granular-ball. It is subsequently partitioned into multiple coarse-grained granular-balls, each of which is iteratively split into finer-grained granular-balls. At splitting step $t$, the coarse-grained granular-ball $\widetilde{\mathcal{G}^t}$ is divided into a set of finer-grained granular-balls, denoted as $\widetilde{\mathbb{G}}^t$.

Specifically, the 0-th step of the splitting phase is the initialization phase, which aims to divide the original graph $\mathcal{G}$ into $a = \sqrt{N}$ granular-balls based on its topology. First, we select the top $a$ nodes with the highest degrees as the center nodes for the $a$ granular-balls:

$$\mathcal{C} = \{c_0, c_1, \cdots, c_a\}, \tag{4}$$

where $\mathcal{C}$ is the set of center nodes. Then the remaining nodes in $\mathcal{G}$ are assigned to center nodes based on the ratio of similarity to minimum hot distance,

$$\text{Assign}(v) = \{\widetilde{\mathcal{G}}_i \mid \min d(v, c_i), c_i \in \mathcal{C}\}, \tag{5}$$

where $\text{Assign}(v)$ denotes that the node $v$ is assigned to the granular-ball $\widetilde{\mathcal{G}}_i$. $d(v, c_i)$ is the ratio of minimum hop distance to similarity between node $v$ and the center node $c_i$,

$$d(v, c_i) = \log \frac{\text{dist}(v, c_i)}{\cos(\mathbf{x}_v, \mathbf{x}_{c_i})}. \tag{6}$$

This results in the initial granular-ball distribution $\widetilde{\mathbb{G}}_{init}$, which covers all nodes in the $\mathcal{G}$. This method quickly achieves a uniform granular-ball distribution while keeping the connectivity of $\widetilde{\mathcal{G}}_i^0$. After the **initialization**, each $\widetilde{\mathcal{G}}_i^0$ undergoes **iterative splitting**.

**Iterative splitting**: At the $t$-th step of the splitting phase, given a granular-ball $\widetilde{\mathcal{G}}_i^t$, the objective is to split $\widetilde{\mathcal{G}}_i^t$ into two finer-grain granular-balls $\widetilde{\mathcal{G}}_1^{t+1}$ and $\widetilde{\mathcal{G}}_2^{t+1}$ for the next iteration.

Specifically, we select the two highest-degree nodes in $\widetilde{\mathcal{G}}_i^t$ as the center nodes. After selecting the center nodes, the remaining nodes in the $\widetilde{\mathcal{G}}_i^t$ are assigned to the center nodes using Eq.5.

Once the nodes in $\widetilde{\mathcal{G}}_i^t$ are assigned, two new granular-balls, $\widetilde{\mathcal{G}}_1^{t+1}$ and $\widetilde{\mathcal{G}}_2^{t+1}$, are formed. Their quality is computed using Eq.3. The splitting procedure ends when Eq.2 is satisfied.

After the iterative splitting phase is complete, each granular-ball in $\widetilde{\mathbb{G}}^t$ is adaptively and iteratively refined into multiple finer-grained granular-balls, resulting in the refined substructures $\{\widetilde{\mathcal{G}}_i\}$ of the original graph $\mathcal{G}$.

**Proposition 2.** *(Proof in Appendix B.1.) The distance function $d(v, c)$ defines a valid metric over the structured domain $(\mathcal{V}, d)$, and the topology it induces supports the Granular-ball covering introduced in Definition 1.*

**Proposition 3.** *(Proof in Appendix B.2.) The Granular-ball covering $\widetilde{\mathbb{G}}$ constitutes a finite topological covering of $\mathcal{V}$ in the metric topology induced by $d$, satisfying: full coverage, local coherence, and neighborhood basis property.*

## 3.3 MULTI-GRANULARITY GRAPH REWIRING

After generating refined granular-balls $\widetilde{\mathbb{G}}$, we discuss how to leverage $\widetilde{\mathbb{G}}$ for graph rewiring. **Multi-Granularity Graph Rewiring** aims to remove low-homophily edges between granular-balls while enhancing homophily within them. It consists of two main processes: Intra-Granular-ball Edge Adding and Inter-Granular-ball Edge Cutting.

The **Intra-Granular-ball Edge Adding** aims to enhance homophily within granular-ball. We add edges to the unconnected node pairs for each $\widetilde{\mathcal{G}}_i$. To ensure the quality of the added edges, we select the top $\delta$ most similar node pairs. Thus the set of added edges is:

$$\mathcal{E}_i^{\text{add}} = \{(v_j, v_k) \mid \text{Top}_{\text{K}}^{\max}(\cos(\mathbf{x}_i, \mathbf{x}_j)), (v_j, v_k) \notin \mathcal{E}_{\widetilde{\mathcal{G}}}\}, \tag{7}$$

where $\text{Top}_{\text{K}}^{\max}(\cdot)$ indicates the selection of the K largest elements and $\text{K} = \delta \cdot |\mathcal{V}_{\widetilde{\mathcal{G}}_i}|$. The set of added edges in graph $\mathcal{G}$ is the union of the added edge sets from all granular-balls $\widetilde{\mathbb{G}}$:

$$\mathcal{E}^{\text{add}} = \bigcup \mathcal{E}_i^{\text{add}}, \tag{8}$$

here, $\bigcup$ denotes the union operation, and $\mathcal{E}^{add}$ is the final edge set after merging edges from all $\widetilde{\mathcal{G}}$.

The **Inter-Granular-ball Edge Cutting** aims to remove low-homophily connections among different $\widetilde{\mathcal{G}}$. To achieve this, we identify the bottom $\gamma$ percentage of edges connecting the least similar node pairs as the edges to be cut:

$$\mathcal{E}^{\text{cut}} = \{(v_m, v_n) \mid \text{Top}_{\text{J}}^{\min}(\cos(\mathbf{x}_m, \mathbf{x}_n)), (v_m, v_n) \in \mathcal{E}\}, \tag{9}$$

where $\text{Top}_{\text{J}}^{\min}(\cdot)$ indicates the selection of the J smallest elements and $\text{J} = \gamma \cdot |\mathcal{E}|$.

Thus, the remaining edge set is:

$$\mathcal{E}^* = \mathcal{E} + \mathcal{E}^{add} - \mathcal{E}^{cut}. \tag{10}$$

### 3.4 MULTI-TASK HOMOPHILY REFINEMENT LEARNING

The **Multi-Task Homophily Refinement Learning** module enables the iterative optimization of the graph rewiring module and the graph neural network (GNN). This process allows the graph rewiring module to fully refine the graph structure while optimizing with the GNN to enhance clustering performance. This section starts with the GNN model.

Our GNN model combines linear transformation and neighborhood aggregation following the prior works Cui et al. (2020a); Dong et al. (2021), which can be expressed as:

$$\mathbf{Z} = (\mathbf{I} - \kappa\mathbf{L})^l \mathbf{X}\mathbf{W}, \tag{11}$$

where $\mathbf{Z}$ denotes the output embeddings, $\kappa$ controls whether the filter is low-pass Cui et al. (2020b), $l$ is the number of propagation times, $\mathbf{X}$ is the feature matrix of the input graph $\mathcal{G}$, and the $\mathbf{W}$ is the weight matrix of linear transformation. $\mathbf{L}$ is the normalized Laplacian matrix, commonly used in GNNs for information propagation and aggregation. It can be defined as:

$$\mathbf{L} = \mathbf{I} - \mathbf{D}^{-\frac{1}{2}}(\mathbf{A} + \mathbf{I})\mathbf{D}^{-\frac{1}{2}}, \tag{12}$$

where $\mathbf{I}$ is the identity matrix, $\mathbf{A}$ is the adjacency matrix of the $\mathcal{G}$, and $\mathbf{D}$ is the degree matrix of $\mathbf{A}$.

A commonly used deep clustering strategy is training the GNN in a self-supervised manner by minimizing the KL divergence:

$$\mathcal{L}_{kl} = \text{KL}(P \mid Q) = \sum_i \sum_k p_{i,k} \log \frac{p_{i,k}}{q_{i,k}}, \tag{13}$$

where the similarity $q_{i,k} \in Q$ can be interpreted as a soft assignment indicating the similarity between the embedding $z_i$ of the node $v_i$ and the $k$-th cluster center $u_k$, representing the probability of $v_i$ being assigned to cluster $k$. It is measured using the Student's t-distribution:

$$q_{i,k} = \frac{(1 + \|\mathbf{z}_i - \boldsymbol{\mu}_k\|^2)^{-1}}{\sum_k^K (1 + \|\mathbf{z}_i - \boldsymbol{\mu}_k\|^2)^{-1}}. \tag{14}$$

To avoid the issue of collapsing into a single cluster caused by directly minimizing the KL divergence, we introduce an auxiliary target distribution $P$ to optimize the clustering distribution:

$$p_{i,k} = \frac{q_{i,k}^2 / \sum_j q_{j,k}}{\sum_k^K \left(q_{i,k}^2 / \sum_j q_{j,k}\right)}, \tag{15}$$

where $0 \leq p_{i,k} \leq 1$ and $p_{i,k} \in P$.

Next, we detail the iterative optimization process between the graph rewiring module and the GNN. During the $i$-th iteration of learning, the rewired graph $\mathcal{G}_i^*$ is used as input to the GNN for training. Then, to ensure the GNN captures the rewired graph $\mathcal{G}_i^*$ structure, we define our graph rewriting objective function:

$$\mathcal{L}_{rw} = \|\mathbf{Z}_i \mathbf{Z}_i^T - \mathbf{A}_i^*\|_F^2. \tag{16}$$

Combining Eq.16 and Eq.13, the overall objective function of GNN can be expressed as:

$$\mathcal{L} = \lambda_{kl} \cdot \mathcal{L}_{kl} + \lambda_{rw} \cdot \mathcal{L}_{rw}, \tag{17}$$

where $\lambda_{kl} > 0$ and $\lambda_{rw} > 0$ are the trade-off parameters. Finally, the rewired graph $\mathcal{G}_i^*$ and the embeddings $\mathbf{Z}_i$ are used for the next iteration of learning.

## 4 EXPERIMENTS

In this section, we outline the experiments conducted to assess the effectiveness of our AGGR framework and address the following research questions: **RQ1:** How does the proposed AGGR method compare to state-of-the-art graph clustering methods in terms of performance? **RQ2:** How do the components of AGGR impact its performance? **RQ3:** How do hyperparameters affect the performance of AGGR? **RQ4:** How robust is AGGR to noise?

### 4.1 DATASETS

In our experiments, we utilized four citation network datasets: Cora, Citeseer, PubmedSen et al. (2008), and ACM, as well as three social network datasets: BlogCatalogTang & Liu (2009), FlickrTang & Liu (2009), and RedditHamilton et al. (2017). BlogCatalog and Flickr are low-homophily, while the others are high-homophily. The detailed statistics of these datasets are provided in Table 4.

### 4.2 BASELINES

We compare the proposed AGGR method with a range of classical clustering algorithms. Specifically, DGI Velickovic et al. (2019) encodes nodes with GNN and then performs the traditional clustering algorithm over the learned embeddings. Besides, three classical deep graph clustering methods DAEGC Wang et al. (2019), SDCN Bo et al. (2020) and AGCN Peng et al. (2021) utilize the graph autoencoder to learn the node representation for clustering. In addition, two representative graph rewiring clustering methods, SUBLIME Liu et al. (2022b) and Hole Gu et al. (2023), optimize the graph structure through graph rewiring, use GNNs to learn the rewired graph, and then perform clustering algorithm on the learned embeddings. Moreover, we evaluate the clustering performance against four state-of-the-art deep graph clustering methods, including AGC Zhang et al. (2019), DCRN Liu et al. (2022c), GCC Fettal et al. (2022), DGCluster Bhowmick et al. (2024) and FPGC Xie et al. (2025).

### 4.3 EXPERIMENTAL SETTINGS

All results are obtained on a Linux server with an NVIDIA RTX 4090 GPU (24GB memory)

**Metrics.** To evaluate the performance of the clustering, we use three widely adopted metrics: Adjusted Rand Index (ARI), Normalized Mutual Information (NMI), and Accuracy (ACC) Liu et al. (2022c); Bo et al. (2020); Tu et al. (2021). Higher values of these metrics indicate better clustering performance. Each experiment is repeated 10 times with different random seeds, and the mean results are reported to ensure robustness. OOM means out of memory.

### 4.4 PERFORMANCE AND ANALYSIS(RQ1)

In Table 1 we report the clustering performance of all compared methods across seven datasets. From the results, we have the following observations: **1) AGGR's superior performance:** The

Table 1: Overall graph clustering performance on seven datasets, measured by ARI, NMI, and ACC (in percentage), is statistically significant with a p-value ¡ 0.01.

| Method | Cora | | | Citeseer | | | Pubmed | | | ACM | | | BlogCatalog | | | Flickr | | | Reddit | | |
|---|---|---|---|---|---|---|---|---|---|---|---|---|---|---|---|---|---|---|---|---|---|
| | ARI | NMI | ACC | ARI | NMI | ACC | ARI | NMI | ACC | ARI | NMI | ACC | ARI | NMI | ACC | ARI | NMI | ACC | ARI | NMI | ACC |
| DGI | 50.77 | 56.63 | 71.54 | 45.49 | 44.63 | 69.27 | 26.70 | 28.96 | 66.02 | 74.71 | 69.67 | 90.72 | 14.57 | 24.32 | 38.33 | 0.60 | 1.31 | 14.59 | OOM | OOM | OOM |
| AGC | 44.76 | 53.49 | 68.91 | 41.60 | 41.00 | 66.99 | 31.71 | 31.51 | 70.16 | 51.23 | 49.58 | 79.93 | 3.12 | 9.82 | 27.70 | 3.76 | 9.55 | 22.35 | OOM | OOM | OOM |
| DAEGC | 44.19 | 49.10 | 67.36 | 41.73 | 41.40 | 66.74 | OOM | OOM | OOM | 70.71 | 65.91 | 88.92 | 3.16 | 4.45 | 25.44 | 2.89 | 5.44 | 19.35 | OOM | OOM | OOM |
| SDCN | 26.85 | 33.25 | 52.41 | 24.10 | 26.26 | 52.30 | 18.50 | 20.75 | 59.80 | 56.41 | 53.08 | 80.64 | 4.04 | 9.21 | 26.43 | 2.10 | 4.88 | 17.98 | 16.12 | 39.50 | 28.04 |
| AGCN | 24.26 | 32.69 | 51.33 | 25.64 | 28.92 | 51.58 | 23.78 | 26.60 | 63.81 | 45.88 | 45.70 | 70.87 | 9.20 | 17.50 | 32.43 | 0.68 | 2.58 | 13.39 | 52.79 | 68.73 | 57.76 |
| DCRN | 24.22 | 39.00 | 54.49 | 44.76 | 42.70 | 68.63 | 33.02 | 33.18 | 70.21 | 69.72 | 64.38 | 88.83 | 2.68 | 4.62 | 24.47 | 1.40 | 2.10 | 15.36 | OOM | OOM | OOM |
| GCC | 51.20 | 58.90 | 74.20 | 46.63 | 45.26 | 70.55 | 33.24 | 32.29 | 70.82 | 47.70 | 54.31 | 65.26 | 21.20 | 34.61 | 50.90 | 3.52 | 22.88 | 24.80 | 43.57 | 68.88 | 56.20 |
| SUBLIME | 46.59 | 51.30 | 67.25 | 42.41 | 43.12 | 67.06 | OOM | OOM | OOM | 64.05 | 60.52 | 85.75 | 45.96 | 63.72 | 75.98 | 28.34 | 58.86 | 64.22 | OOM | OOM | OOM |
| DGCluster | 43.55 | 55.18 | 68.13 | 28.11 | 42.17 | 38.34 | 34.74 | 35.82 | 66.72 | 72.13 | 70.69 | 82.16 | 20.07 | 26.78 | 44.02 | 8.07 | 12.74 | 25.64 | OOM | OOM | OOM |
| HoLe | 53.91 | 57.63 | 74.33 | 48.19 | 45.24 | 72.38 | 33.75 | 31.75 | 71.34 | 80.10 | 74.57 | 92.90 | 81.07 | 78.43 | 91.49 | 60.59 | 62.49 | 75.40 | 66.23 | 77.51 | 70.95 |
| FPGC | 59.11 | 59.55 | 79.19 | 46.61 | 46.36 | 72.59 | 31.34 | 34.57 | 71.03 | 70.96 | 64.74 | 88.85 | 35.55 | 18.95 | 10.29 | 22.62 | 10.41 | 4.98 | OOM | OOM | OOM |
| AGGR | 63.22 | 62.32 | 79.56 | 50.61 | 48.43 | 73.56 | 43.46 | 37.71 | 76.74 | 80.28 | 74.99 | 92.96 | 83.96 | 81.04 | 92.88 | 66.22 | 67.33 | 82.01 | 68.95 | 77.92 | 73.44 |

proposed AGGR method consistently outperforms all baselines across seven datasets and evaluation metrics, with particularly strong improvements on low-homophily social networks. This improvement stems from AGGR's granular-ball graph refinement method, which adaptively refines the graph into highly homophily substructures and enhances homophily through graph rewiring, thereby improving clustering performance. **2) Advantages over fine-grained methods:** SUBLIME, Hole, and AGGR enhance graph quality through graph rewiring, achieving good performance, particularly on heterophily graphs. Compared to fine-grained methods SUBLIME and Hole, AGGR adopts a coarse-grained approach that extracts homophily substructures. This reduces the impact of graph noise on graph rewiring, improves graph homophily more effectively, and enhances performance.

Table 2: The efficacy of different modules within the AGGR framework.

| Modules | | Cora | | | Flickr | | |
|---|---|---|---|---|---|---|---|
| | | ARI | NMI | ACC | ARI | NMI | ACC |
| w/o | MGGR | 50.23 | 55.42 | 71.64 | 41.38 | 48.98 | 65.53 |
| w/o | MHRL | 56.01 | 58.33 | 74.37 | 51.52 | 53.58 | 71.96 |
| w/ | AGGR | 63.22 | 62.32 | 79.56 | 66.22 | 67.33 | 82.01 |

## 4.5 Ablation Studies(RQ2)

To address RQ2, we conducted two ablation studies for AGGR. The first ablation study aims to evaluate the effectiveness of each module in AGGR. Table 2 presents the results. **(1) Without the Multi-Granularity Graph Rewiring component**, the resulting variant (w/o MGGR) includes only the GNN module, leading to a notable performance decline across all datasets and evaluation metrics. This observation underscores the importance of high-quality input graphs in graph clustering tasks. It also highlights the significance of Multi-Granularity Graph Rewiring module in refining the graph. **(2) Impact of Multi-Task Homophily Refinement Learning (MHRL):** The inclusion of the MHRL module significantly enhances performance. This highlights the MHRL module's effectiveness in optimizing graph rewiring and GNNs synergistically.

We also conducted related ablation experiments to investigate the effects of edge addition and removal operations, with specific experimental results provided in Appendix C.

## 4.6 Hyperparameter Analysis(RQ3)

We conducted a detailed analysis of the hyperparameters $\delta$ and $\gamma$, which control the proportion of added edges within granular regions and the proportion of deleted edges between granular regions, respectively. For $\delta$ and $\gamma$, we evaluate five typical values: $\{0.2, 0.4, 0.6, 0.8, 1.0\}$. As shown in Fig.3, the effects of individual hyperparameters on clustering performance demonstrate diverse change trends across different datasets. On the high-homophily Pubmed dataset, performance is influenced by $\delta$ but less affected by $\gamma$. Conversely, on the low-homophily Flickr dataset, performance is less sensitive to $\delta$ but more impacted by $\gamma$. This can be attributed to the following reasons: In

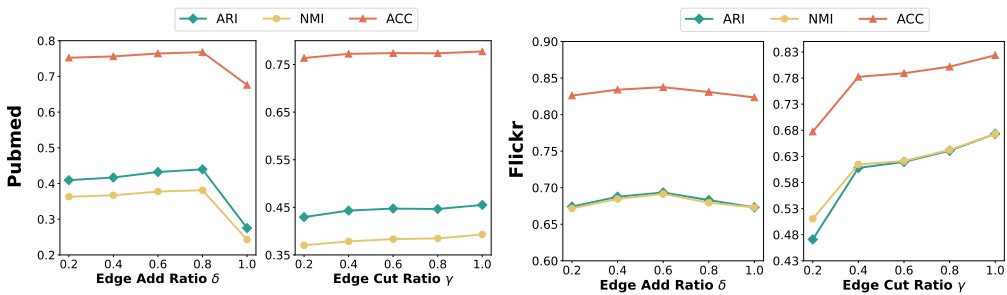

Figure 3: Parameters Analysis of $\delta$ and $\gamma$.

high-homophily graphs, more homophily edges allow AGGR to form higher-quality homophily subdomains. Adding edges effectively improves graph homophily, while edge removal has limited impact. Additionally, when $\delta$ is too large, the quality of added edges cannot be ensured, leading to performance degradation. In contrast, in low-homophily graphs, removing heterophily edges effectively enhances graph homophily, improving performance.

Table 3: Evaluation Results under Different Noise Rates $r$.

| Dataset | $r$ | AGGR | | | Hole | | |
|---|---|---|---|---|---|---|---|
| | | ARI | NMI | ACC | ARI | NMI | ACC |
| Citeseer | 0.05 | 49.54 | 47.33 | 72.28 | 45.92 | 44.51 | 69.58 |
| | 0.1 | 48.79 | 46.62 | 71.64 | 45.62 | 44.29 | 69.37 |
| | 0.15 | 47.93 | 45.78 | 71.06 | 45.42 | 44.23 | 69.10 |
| | 0.2 | 46.72 | 44.25 | 70.33 | 43.87 | 43.02 | 68.17 |
| Flickr | 0.05 | 65.19 | 66.49 | 81.15 | 54.31 | 56.64 | 72.48 |
| | 0.1 | 64.27 | 65.41 | 80.65 | 52.38 | 54.28 | 71.39 |
| | 0.15 | 64.01 | 65.27 | 80.20 | 51.80 | 54.14 | 70.36 |
| | 0.2 | 63.02 | 64.45 | 79.43 | 51.85 | 53.70 | 69.78 |

### 4.7 ROBUSTNESS ANALYSIS UNDER NOISE(RQ4)

To evaluate the noise robustness of AGGR, we conducted noise experiments. The experimental setup is as follows: To add noise without compromising the graph's homophily, for the high-homophily Citeseer dataset, we introduce noise by randomly deleting edges in the graph. In contrast, for the low-homophily Flickr dataset, we introduce noise by randomly adding edges. As shown in Table 3, we evaluated the performance of AGGR and Hole Gu et al. (2023) under 5%, 10%, 15%, and 20% noise levels on the Citeseer and Flickr datasets. AGGR maintains high performance on Citeseer and Flickr. AGGR's noise robustness stems from its multi-granularity granular-ball computation, which conducts graph rewiring operations on homophily regions extracted from the graph based on homophily quality, effectively reducing the impact of noise on graph rewiring performance.

## 5 CONCLUSION

This paper introduces a novel graph clustering method, Adaptive Granularity Graph Rewiring via Granular-Ball for Graph Clustering (AGGR). The core innovation of AGGR lies in incorporating the multi-granularity distribution of homophily into the graph rewiring process, enabling an adaptive granular approach. This method adaptively refines the graph's homophilic structure, subtly enhancing homophily and thereby improving clustering performance. Furthermore, the Multi-Task Homophily Refinement Learning module is developed to effectively utilize clustering information, further advancing the performance of both graph rewiring and graph neural networks. We conduct extensive experiments on seven real-world datasets, demonstrating AGGR's superiority. However, AGGR currently focuses on static graph clustering, with future work aiming to extend it to temporal graph.

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

# A DATASET

Table 4: Dataset Statistics. Hom. is short for homophily.

| Dataset | Node | Edge | Feat | Cluster | Hom. |
|---|---|---|---|---|---|
| **Cora** | 2,708 | 5,429 | 1,433 | 7 | 0.81 |
| **Citeseer** | 3,327 | 4,732 | 3,703 | 6 | 0.74 |
| **ACM** | 3,025 | 8,117 | 13,128 | 3 | 0.82 |
| **Pubmed** | 19,717 | 88,651 | 500 | 3 | 0.80 |
| **BlogCatalog** | 5,196 | 343,486 | 8,189 | 6 | 0.40 |
| **Flickr** | 7,575 | 479,476 | 12,047 | 9 | 0.24 |
| **Reddit** | 232,965 | 114,615,892 | 602 | 41 | 0.76 |

# B PROOFS

## B.1 PROOFS REGARDING PROPOSITION 2

*Proof.* We verify the three standard conditions required for $d$ to define a metric:

- Non-negativity: The shortest-path distance $dist(v, c)$ is always at least 0, and the cosine similarity $\cos(\mathbf{x}_v, \mathbf{x}_c)$ lies in the interval $(0, 1]$. Therefore, the ratio inside the logarithm is greater than or equal to 1, ensuring that $d(v, c) \geq 0$.

- Symmetry: Both $dist(v, c)$ and $\cos(\mathbf{x}_v, \mathbf{x}_c)$ are symmetric with respect to their inputs. Hence, $d(v, c) = d(c, v)$ holds for all node pairs.

- Triangle inequality: Since $dist(v, c)$ satisfies the triangle inequality and cosine similarity varies smoothly under local feature consistency assumptions, the combined formulation of $d(\cdot, \cdot)$ preserves the triangle inequality. More precisely, for nodes in connected neighborhoods, the variation in cosine similarity is moderate, and thus the logarithmic form ensures that the additive triangle inequality $d(v, w) \leq d(v, u) + d(u, w)$ remains satisfied.

Consequently, $d(\cdot, \cdot)$ defines a valid metric, and the topology induced by this distance function is suitable for constructing closed Granular-balls $\widetilde{\mathcal{G}}_i = \{v \in \mathcal{V} \mid d(v, c_i) \leq r_{c_i}\}$. These balls serve as basic neighborhoods, and the collection $\tilde{\mathbb{G}}$ forms a finite topological covering of $\mathcal{V}$, which supports localized, scalable, and topology-aware processing in the Granular-ball Computing framework. □

## B.2 PROOFS REGARDING PROPOSITION 3

*Proof.* The proof is structured around verifying the three topological properties within the metric topology $(\mathcal{V}, \mathcal{T}_d)$:

- Coverage: By construction, each Granular-ball $\widetilde{\mathcal{G}}_i$ is defined as a closed metric ball $\{v \in \mathcal{V} \mid d(v, c_i) \leq r_{c_i}\}$ centered at $c_i$ with radius $r_{c_i} > 0$. The set $\tilde{\mathbb{G}}$ is chosen such that:

$$\bigcup_{i=1}^{n} \tilde{\mathcal{G}}_i \supseteq \mathcal{V}, \tag{18}$$

ensuring that every node $v \in \mathcal{V}$ belongs to at least one Granular-ball, thereby achieving complete topological coverage.

- Locality: Since $d$ is a valid metric on $\mathcal{V}$, every point $v \in \mathcal{V}$ admits a neighborhood basis composed of open balls in the topology $\mathcal{T}_d$. Although $\widetilde{\mathcal{G}}_i$ is defined as a closed ball, it still includes all points within the specified radius and is thus a closed neighborhood, which can approximate any open neighborhood arbitrarily well in discrete metric spaces such as graphs. Therefore, each $\widetilde{\mathcal{G}}_i$ maintains local coherence and supports localized computations.

Table 5: The comparison of Time and Space between AGGR and Hole.

| Method | Metric | Cora | Flickr | Pubmed | Reddit |
|--------|--------|------|--------|--------|--------|
| **HoLe** | Per-epoch time (s) | 0.0299 | **0.08603** | **0.15657** | **20.54797** |
| | Running time (s) | 8.5735 | 74.9463 | 112.1944 | 963.9197 |
| | GPU peak memory (MB) | 430.89 | 2634.78 | 10837.61 | 6376.13 |
| **AGGR** | Per-epoch time (s) | **0.0128** | 0.0966 | 0.1772 | 30.43237 |
| | Running time (s) | **5.1729** | **65.3561** | **79.2644** | **835.68812** |
| | GPU peak memory (MB) | 466.52 | 3118.26 | 11436.47 | 9397.03 |

- Basis Neighborhood. In metric-induced topologies, closed balls form a base of closed neighborhoods. As $\tilde{\mathbb{G}}$ is a finite collection of such balls with adjustable centers and radii, it satisfies the basis property that for every $v \in \mathcal{V}$ and every neighborhood $U$ of $v$ in $\mathcal{T}_d$, there exists some $\widetilde{\mathcal{G}}_i$ such that $v \in \widetilde{\mathcal{G}}_i \subseteq U$.

Together, these three properties ensure that the Granular-ball covering $\tilde{\mathbb{G}}$ not only respects the metric topology but also enables a scalable and topology-aware decomposition of the graph domain. This provides a rigorous foundation for building efficient graph algorithms within localized topological structures. $\qquad\square$

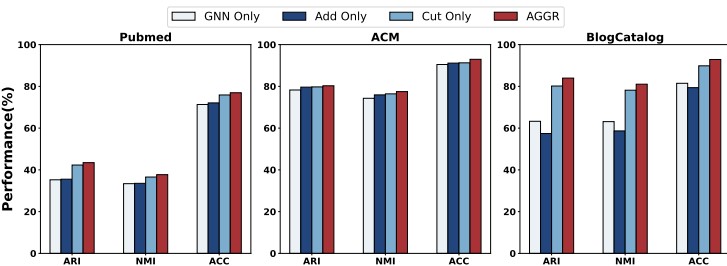

Figure 4: A comparison of different rewiring operations.

## C    DIFFERENT OPERATIONS ANALYSIS

Specifically, the Cut-only variant performs only edge removal during the graph rewired process, while the Add-only variant focuses solely on edge addition. As shown in Fig. 4, both edge addition and edge removal are essential for AGGR. Both operations improve performance on high-homophily graphs, such as Pubmed and ACM. However, on the low-homophily graph BlogCatalog, where noise is more prevalent, edge addition leads to performance degradation, while edge removal proves to be a more critical operation, resulting in performance improvements.

## D    COMPLEXITY ANALYSIS

We have conducted a detailed analysis of AGGR's computational complexity. The overall time complexity of Granular-ball graph refinement is $\mathcal{O}(N^{3/2} + M\sqrt{N})$ where $N$ is the number of nodes and $M$ is the number of edges. This complexity mainly comes from the granule construction and aggregation process, ensuring scalability for large graphs.

For space complexity, AGGR requires memory for: 1.graph structure storage $O(N + M)$, 2.node feature storage $O(Nd)$, 3.Granular-ball center node storage $O(\sqrt{N}d)$, 4.Granular-ball index for each node $O(N)$, 5.model parameters storage $O(P)$.

Thus, the total space complexity is $O(N + M + Nd + \sqrt{N}d + P)$, where $d$ is the feature dimension and $P$ is the number of learnable parameters in the GNN model. This is manageable for practical applications.

We compared AGGR with the baseline (HoLe) on three benchmark datasets: Cora, Flickr, Pubmed, and Reddit. The results are as follows:

# E    VISUALIZATION

To intuitively validate the effectiveness of our method, we utilized the $t$-SNE Van Der Maaten (2014) algorithm to create 2D clustering visualization plots. As shown in Fig.5, the feature representations generated by our method exhibit the best separability, with samples of the same class clustered closely together and distinct gaps between different classes. This shows that AGGR produces the most discriminative representations of characteristics.

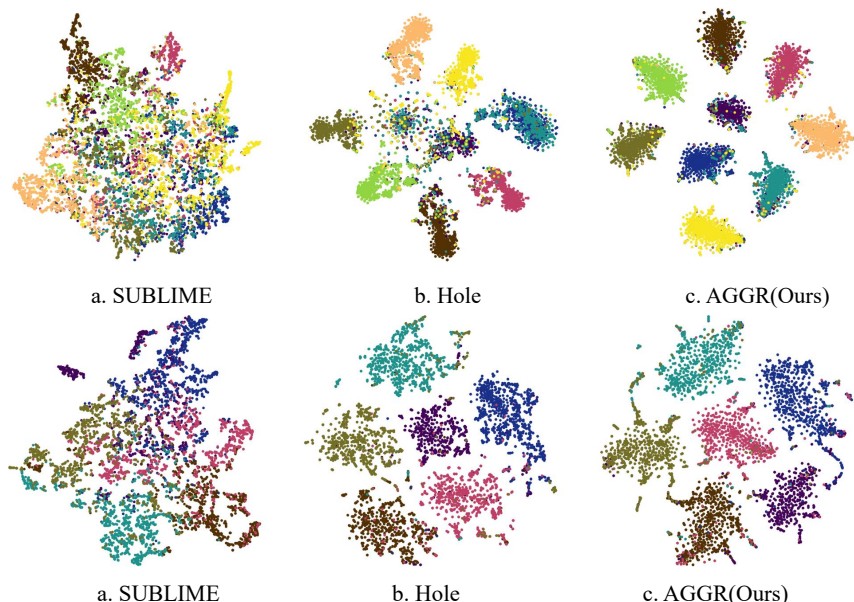

Figure 5: Visualization of different models on Flickr(1st row) and BlogCatalog(2nd row) $t$-SNE.

