# OpenReview forum: "Adaptive Granularity Graph Rewiring via Granular-ball for Graph Clustering"
_ICLR.cc/2026/Conference — Submitted to ICLR 2026_

### Official Review · Reviewer_nWxp · 2025-10-27

**Soundness:** 2
**Presentation:** 3
**Contribution:** 2
**Rating:** 4
**Confidence:** 4

**Summary:**

This paper introduces a novel graph clustering method, Adaptive Granularity Graph Rewiring via Granular-Ball for Graph Clustering (AGGR). The core innovation of AGGR lies in incorporating the multi-granularity distribution of homophily into the graph rewiring process, enabling an adaptive granular approach. This method adaptively refines the graph’s homophilic structure, subtly enhancing homophily and thereby improving clustering performance. Furthermore, the Multi-Task Homophily Refinement Learning module is developed to effectively utilize clustering information, further advancing the performance of both graph rewiring and graph neural networks. Experimental results indicate that AGGR demonstrates competitive clustering performance metrics across multiple datasets in low-homogeneity social networks compared to existing methods.

**Strengths:**

1. The paper introduces a novel graph clustering method, Adaptive Granularity Graph Rewiring (AGGR), which optimizes graph structures by identifying and rewiring multi-granularity homogeneous regions.

2. The paper clearly explains the theoretical foundation and implementation details of the AGGR method, including the identification of multi-granularity homogeneity distribution and the application of adaptive granular-ball mechanisms.

**Weaknesses:**

1. The paper does not explain the related work on graph rewiring, making it difficult to assess the novelty of the proposed graph rewiring method.

2. The experiments for RQ2 and RQ4 were conducted on only two datasets, which is not sufficiently convincing.

3. RQ1 should be compared with other graph rewiring methods to demonstrate its superiority over existing approaches.

**Questions:**

1. What are the specific distributions P and Q in the KL divergence loss presented in Equation (13)?
2. The paper appears to apply a graph rerouting method to graph clustering, but it is unclear what the necessary conditions and motivations are for using graph rerouting in graph clustering.
3. What are the innovations of this paper compared to other graph rerouting methods?

---

### Official Review · Reviewer_SANR · 2025-11-01

**Soundness:** 2
**Presentation:** 2
**Contribution:** 2
**Rating:** 2
**Confidence:** 4

**Summary:**

This paper proposes a graph modification algorithm named AGGR, designed for graph clustering tasks. The paper emphasizes the multi-granular nature of graphs, suggesting that homophily is distributed across multiple granular levels, and that homophilic nodes should be distinguished at the optimal granularity. Based on this idea, the authors introduce an adaptive granular graph rewiring method. AGGR can adaptively identify homophilic regions at appropriate granularities. It introduces a graph refinement mechanism based on adaptive granular spheres and employs a multi-granular graph rewiring strategy. Experimental results demonstrate that the proposed approach achieves strong performance.

**Strengths:**

The methodology of the paper is clearly presented and easy to follow. Moreover, the proposed AGGR method seems to perform well in the experiments.

**Weaknesses:**

1. It is unclear to me how the multi-granular nature of graphs benefits the clustering task. This aspect appears to be central to the paper, yet unfortunately it is only briefly discussed in the introduction. A more thorough theoretical and empirical analysis is expected.

2. Some of the claims in the paper, such as those concerning the relationship between graph granularity and homophily, lack theoretical justification, empirical validation on real datasets, or supporting references. As a result, these statements are not fully convincing.

3. Moreover, I am not convinced that such multi-granular properties are widely present in real-world graph data.

**Questions:**

1. Is it necessary for us to introduce two lambdas in Eq. 17, or would setting one of them to $1$ already be sufficient, thus saving a hyperparameter?

2. In Section 3.3, why is the ratio for adding edges related to the number of nodes, expressed as $\delta * |V|$, while the ratio for deleting edges is related to the number of edges, expressed as $\gamma * |E|$?

3. In Section 4.6, does the setting where $\gamma$ is set to 1 mean that all edges are removed? That seems unreasonable — please clarify. Do the experimental results suggest that we should always choose the largest possible $\gamma$?

---

### Official Review · Reviewer_cCXX · 2025-11-01

**Soundness:** 2
**Presentation:** 3
**Contribution:** 2
**Rating:** 4
**Confidence:** 3

**Summary:**

The paper proposes MGGR for graph classification, focusing on multi-granularity subdomains via granular-ball decomposition. It adaptively partitions graphs ($\sqrt N$ initialization, recursive binary splits) using a quality criterion that combines structural measures and label purity, then applies a hierarchical encoder with intra-domain structural feature aggregation and inter-domain GNN modeling. Experiments on several benchmarks show consistent accuracy gains over strong baselines and robustness to label noise.

**Strengths:**

- Interpretable multi-granularity decomposition that preserves intra- and inter-subdomain structure for finer, structure-aware representations.

- Effective hierarchical encoding (intra-domain structural features + inter-domain GNN) capturing local/global patterns, with strong accuracy and noise robustness.

- Adaptive splitting with quality criteria and local computations improves scalability; validated by comprehensive ablations and sensitivity analyses.

**Weaknesses:**

- Dependence on label purity in splitting limits applicability to unsupervised or weakly supervised settings.

- Heuristic partitioning ($\sqrt N$ centers, highest-degree seeds, binary splits) may be sensitive to degree skew/topology, with limited theoretical guarantees.

- Hand-crafted intra-domain structural features are not end-to-end learned, potentially limiting expressiveness and task adaptivity.

- Nontrivial overhead for computing eigen/centrality/diameter; graph-level readout is under-specified and runtime/memory analysis is limited.

- Related work gap: several hierarchical graph representation learning approaches are not sufficiently discussed [1, 2].

[1] Galaxy Network Embedding: A Hierarchical Community Structure Preserving Approach.

[2] Hierarchical community structure preserving network embedding: A subspace approach

**Questions:**

- The splitting criterion relies on label purity. How does MGGR operate when labels are unavailable or highly noisy?

---

### Meta-Review · Area_Chair_A8Ua · 2025-12-31

**Summary:**

In this paper, the authors propose AGGR, an adaptive granularity graph rewiring method for graph clustering based on granular-ball decomposition. While the idea of leveraging multi-granularity homophily is interesting, the contribution is not sufficiently supported by strong theoretical justification or convincing empirical evidence. Reviewers raised concerns about novelty, clarity, evaluation scope, and assumptions behind the method. However, the authors did not provide any rebuttal, and none of the concerns raised by the reviewers were addressed during the discussion phase. Overall, the paper does not meet the acceptance standard.

**Reviewer Concerns:**

The concerns from the reviewers were not addressed, as the authors did not provide a rebuttal.

Multiple reviewers questioned the core motivation and novelty of the proposed approach. In particular, the claim that homophily is distributed across multiple granular levels and that identifying an “optimal granularity” is critical for clustering lacks strong theoretical grounding and convincing empirical validation. Reviewers noted that these claims are central to the paper but are only briefly discussed and not well supported by evidence or prior literature.

There were also concerns about the method design and assumptions, including reliance on label purity during the splitting process, heuristic granular-ball partitioning, and the use of hand-crafted structural features rather than end-to-end learned representations.

**Reviewer Scores:**

All reviewers would remain their score since the authors did not provide a rebuttal.

---

### Decision · Program_Chairs · 2026-01-26

Reject